# Identity Recognition in Sanitary Facilities Using Invisible Electrocardiography

**DOI:** 10.3390/s22114201

**Published:** 2022-05-31

**Authors:** Aline Santos Silva, Miguel Velhote Correia, Francisco de Melo, Hugo Plácido da Silva

**Affiliations:** 1FEUP—Faculdade de Engenharia da Universidade do Porto, 4200-465 Porto, Portugal; mcorreia@fe.up.pt; 2IT—Instituto de Telecomunicações, 1049-001 Lisboa, Portugal; francisco.de.melo@tecnico.ulisboa.pt (F.d.M.); hsilva@lx.it.pt (H.P.d.S.); 3INESC TEC—Institute for Systems and Computer Engineering, Technology and Science, 4200-365 Porto, Portugal; 4Department of Bioengineering, Instituto Superior Técnico, 1049-001 Lisboa, Portugal

**Keywords:** invisibles, off-the-person, electrocardiography, pervasive sensing, telemedicine, identity recognition, biometrics

## Abstract

This article proposes a new method of identity recognition in sanitary facilities based on electrocardiography (ECG) signals. Our team previously proposed a novel approach of invisible ECG at the thighs using polymeric electrodes, leading to the creation of a proof-of-concept system integrated into a toilet seat. In this work, a biometrics pipeline was devised, which tested four different classifiers, varying the population from 2 to 17 subjects and simulating a residential environment. However, for this approach to be industrially viable, further optimization is required, particularly regarding electrode materials that are compatible with industrial processes. As such, we also explore the use of a conductive silicone material as electrodes, aiming at the industrial-scale production of a toilet seat capable of recording ECG data, without the need for body-worn devices. A desirable aspect when using such a system is matching the recorded data with the monitored user, ideally using a minimal sensor set, further reinforcing the relevance of user identification through ECG signals collected at the thighs. Our approach was evaluated against a reference device for a population of 17 healthy and pathological individuals, covering a wide age range (24–70 years). With the silicone composite, we were able to acquire signals in 100% of the sessions, with a mean heart rate deviation between a reference system and our experimental device of 2.82 ± 1.99 beats per minute (BPM). In terms of ECG waveform morphology, the best cases showed a Pearson correlation coefficient of 0.91 ± 0.06. For biometric detection, the best classifier was the Binary Convolutional Neural Network (BCNN), with an accuracy of 100% for a population of up to four individuals.

## 1. Introduction

Biometric identification is the use of anatomical or physiological characteristics to determine the identity of a user before a system. Methods such as fingerprint or facial recognition currently have widespread use in real-world applications, however, research has explored other modalities such as the gait or Electroencephalogram (EEG) [1,2,3]. Cardiovascular signals are readily available and have also been shown to have interesting properties for biometric recognition. In particular, the ECG has been shown to have low intra-subject variability and high inter-subject variability, making it a promising modality for biometrics [2,4].

In [5], we introduced a proof-of-concept (PoC) system for invisible ECG, implemented in the form of a toilet seat. The PoC allowed for us to experimentally demonstrate the possibility of measuring ECG signals on the thighs. A specific sensor was designed, and polymeric dry electrodes with conductive properties were used as the interface with the body. Given the natural barriers to signal propagation (e.g., androgenic hair), this previous work also explored different textures of the electrode surface. While simpler textures (i.e., a flat surface) do not allow for adequate ECG signal acquisition in some morphotypes, a pyramidal texture consistently showed adequate results across all the different criteria under evaluation. This opens up a new approach to automated health-monitoring systems that can function as an extension of people’s everyday lives, without changing the look and feel of the objects in which they are integrated.

Based on our results, an instrumented toilet seat was created, which aggregates the technical solutions that demonstrated the best performance during the research and development process. However, considering that the use of such a device may be shared amongst multiple inhabitants within a household, a desirable property of such a system is the ability to match the recorded data to the subject that provided it. Although previous work has demonstrated the biometric potential of ECG signals, in this work, we investigate the biometric potential of ECG signals acquired at the thighs, which differ significantly from signals acquired at standard locations.

We explored Binary Convolutional Neural Network (BCNN), Support Vector Machines (SVM), Gaussian Naive Bayes (GaussianNB), and the K-Nearest Neighbors algorithm (3-NN) as classifiers to identify different users. In doing so, we tested a different number of registered users to simulate households of different sizes. To our knowledge, this is the first study of its kind.

Furthermore, for the final implementation, two elongated electrodes were designed with a shape consistent with the toilet seat (Figure 1 and Figure 2a). These retain the pyramid-like texture proposed in [5,6], but modified so that the portion of the electrode in contact with the body has a more aesthetically pleasing appearance. Conductive silicone is also explored for the electrode material, which increases the comfort and has a higher compatibility with industrial processes.

The paper is presented as follows. Section 2 describes the background and most relevant previous work in this area. Section 3 presents the main details of the experimental setup. Section 4 summarizes the methods used to identify and classify individuals. Section 5 summarizes the experimental evaluation and results. Finally, Section 6 presents the main conclusions and future work directions.

## 2. Related Work

The accurate and automatic identification of individuals is becoming an integral part of our daily lives. For our use case, automatic identification is needed to distinguish individuals when they use the toilet seat for ECG acquisition; however, there are usability and privacy aspects related with this particular application that make it especially challenging.

The topic of identity recognition in sanitary facilities is extremely rare in the literature. In [7], the authors describe a patent application related to a toilet seat and smart close-stool, designed to monitor human bioimpedance using multiple electro-acoustic transducers applied to the toilet seat body on both sides. To address the problem of user identification, an external unit is proposed to obtain the fingerprint.

The authors of [8] also highlight the relevance of obtaining user identification, in this case in the context of a multifunctional toilet cover plate. In addition to acquiring ECG signals, the authors proposed a keyboard, an iris scanner, or a fingerprint sensor for identity recognition. However, these options exhibit multiple usability constraints when considering the target application.

A recent work by Park et al. [9] describes a smart toilet designed for the long-term analysis of excreta through multimodal data collection. In this work, the problem of user identification is addressed through fingerprint recognition, but a novel approach is also proposed, based on anodermic features extracted from video imaging.

The authors of Zhang et al. [10] describes an Artificial Intelligence toilet (AI-Toilet) for an integrated health-monitoring system (IHMS) using intelligent triboelectric pressure and image sensors. This study addresses a frustum structure and the spacer structure in the eco-flex layer to extend the detection range for measuring the seating pressure. The biometric information obtained with this prototype from six users sitting on the toilet seat enabled identification with 97.14% accuracy.

Kurahashi et al. [11] focused on another perspective in their work, by focusing on the creation of a personal identification system based on the rotation of toilet paper rolls. They focus on the differences in the way a toilet roll is pulled, proposing a system that identifies individuals based on the rotation characteristics of toilet rolls with a gyroscope. A recognition accuracy of 83.9% was reached for a group of five individuals.

If we can demonstrate the feasibility of thighs ECG signal as biometric, then the recognition system using the ECG can be activated as an extension of the regular monitoring process. However, due to the characteristics of the thighs ECG, there are variability conditions that greatly affect the signal and its quality [2,12,13].

## 3. Experimental Setup

To test our proposed approach, a full-scale toilet seat model was created, as in Figure 1, with an indentation on each side to slot the electrodes. The electrical connection to the sensor module located on the back is made by means of a wire soldered to a brass plate, which, in turn, is capped by the electrode. The sensor and data acquisition system is the same as that used in [5], allowing for the toilet seat to transmit the signals collected via Bluetooth to a receiver (i.e., a computer or smartphone).

### 3.1. Conductive Silicone

A central component of our work is the interface with the body, using non-gelled materials that can be used as an electrode. In [5], we explored the Proto-Pasta CDP1 (ProtoPlant Inc., Vancouver, WA, USA) conductive Polylactic Acid (PLA) carbon black/polymer thermoplastic [14], herein described as a PLA Electrode. Experimental results for this material on a 1 cm^3^ sample have shown a surface resistivity between 150 and 200 Ω cm (along the Y axis), and a volume resistivity between 170 and 240 Ω cm (along the Z axis) [14].

However, the melting point (195–215 °C), decomposition (250 °C), and auto-ignite (300 °C) temperatures of the PLA Electrode [14] have proven to be technically challenging for industrial scale-up. Fused filament fabrication (FFF), the process for which this material was primarily designed, also has low resource efficiency and presents long-term mechanical robustness problems. The latter is particularly troublesome when minutiae geometries are involved (as is the case in our hemispherical texture), and when the parts are repeatedly subjected to abrasion (e.g., when the surface is cleaned with a scrub sponge).

While the PLA Electrode will be used for benchmarking with [5], in this work we explore a conductive silicone material as a new alternative. A custom electrode was produced in the shape of the chosen texture, and evaluated through experimental trials. In particular, Conductive High-Consistency Silicone Rubber filled with carbon black is used as the material for the electrode (herein designated as a Silicone Electrode). Vulcanised parts made from this compound have a unique combination of properties, characterised by their good flexibility, mechanical properties, electrical specifications and very good industrial processing properties (suitable for both injection moulding and extrusion applications, Table 1).

### 3.2. Electrode Texture

When measured at the thighs, ECG signals have amplitudes in the μV range [5]. An adequate electrical contact between the skin of the subject and the electrode is, therefore, of paramount importance. However, the electrical contact can be compromised by natural barriers such as androgenic hair [15,16], which is a particularly relevant problem in the context of our work. For benchmarking purposes, we use the PLA Electrode with the hemispherical texture (Figure 2a), as it achieved the best results in [5].

Nevertheless, for the Silicone Electrode, a new texture was designed and tested; this design has slight undulations and longer ridges, which was found to be more aesthetically pleasing and suitable for real-world deployment (Figure 2b).

### 3.3. Skin-to-Electrode Impedance

Some factors can affect the impedance of the skin, resulting in a change in ECG signal quality. For example, moisturisers, age, sunlight, humidity and temperature can affect the impedance of the skin. This impedance is measured as a voltage-to-current ratio and can be described as the resistance experienced by the alternating electrical current at the interface between the skin and the electrode. When the electrode makes contact with the skin surface, the skin becomes part of the measurement circuit. This can result in significant noise and even signal loss as the insulation of the skin–electrode interface degrades the circuit due to high skin impedance, leading to lower signal quality.

A circuit was created with a resistor *R* = 1 kΩ, two pre-gelled Ag/AgCl electrodes, and the experimental electrode, for which we would like to determine the impedance in contact with the skin (Figure 3).

Using the oscilloscope, we determined the maximum amplitude value for each channel, $V_{A}$ and $V_{B}$. From these values, the amplitude of the impedance was determined according to Equation (Equation 1).
(1)$$Z\left(k\Omega \right)=R\;\cdot \;\frac{V_{A}}{V_{B}}$$

Silicone showed similar behaviour to the best-performing electrode, PLA, as shown in Figure 4 and Table 2.

### 3.4. ECG Acquisition Setup

In parallel with the toilet seat, data are simultaneously collected with a second system applied at a reference location on the body of the tested subjects (herein considered as the reference system). The experimental setup also included a luminosity sensor (LUX) on the reference system and a LED on the toilet seat, to enable the synchronization of both independent time series in post-processing. This optical approach was adopted to ensure electrical decoupling between the two systems.

Figure 5 shows the general design of our study. Data acquisition was simultaneously performed with our setup, the reference system, and the silicone electrodes. From our setup, four data sources were produced; these were channels O1—LED and A1—ECG REF in the reference device, and channels A1—ECG EXP and A6—LUX in the toilet seat. The reference (ECG REF) sensor was applied to the subject with the IN+ terminal on the left clavicle, the IN—terminal on the right, and the REF on the cervical—C5/C6 (Figure 5), corresponding to an Einthoven Lead I surrogate.

With the ECG REF, disposable Covidien (Covidien Ltd., Gosport, UK) KENDALL ARBO H124SG EMG/ECG/EKG surface electrodes (24-mm diameter) were used, while the electrodes described in Section 3.2 were used with the toilet seat sensor. The latter is a dry electrode, while the Covidien electrode is a typical clinical use electrode, which has an adhesive side with non-irritating gel, especially developed to improve conductivity, preventing allergic reactions.

## 4. Classification and Identification

Figure 6 shows the pipeline for our system analysis, which was organised in the following order: ECG data acquisition, pre-processing, R-peak detection, feature extraction and, finally, classification for identity recognition.

### 4.1. Data Acquisition

A total of 17 healthy and pathological volunteers aged 24–70 years were enrolled for this phase; of these subjects, 13 were female and 2 had androgenic body hair. For each participant, 5 min of data were recorded while sitting down with the skin in contact with the electrodes and the legs projected to the front (heels touching the ground, to ensure a uniform weight distribution across the electrodes). Data acquisition was performed according to the principles of the Declaration of Helsinki, following an approval of the protocol by the Data Protection Officer (PDO) of the Instituto de Telecomunicações (IT) to ensure that the experimental protocols and methods conformed to the previously established ethical guidelines. Informed consent was obtained from all the participants.

For data analysis, mean and standard deviation were calculated for each subject. Python 3.8 and the BioSPPy (0.6.1) [17] and PyHRV (0.4.0) libraries were used for digital filtering and segmentation methods and heartrate variability (HRV) analysis, respectively. Further details on the signal pre-processing steps can be found in the article [5]. However, the analysed signals were filtered using a Finite Impulse Response (FIR) bandpass filter of order 300 with cuttoff frequency of 3–45 Hz, and were segmented according to the method proposed by Hamilton [15].

### 4.2. Biometric Templates

After extracting the ECG signal characteristics for each subject, an average of 90 templates per subject was obtained. From this template total, two types of analysis were performed, pertaining the selection of the training and testing sets: random and static. For random profiling, the templates of each subject were randomly selected to create the testing and training sets, and these formed the inputs to the classifiers. For the static characterization, the goal was to find out if the performance of the classifier is the same when there is a gap in the timeline. That is, for each subject, the first 30 templates from each subject were chosen for the training and for test sets, and separated into the following proportions:$\frac{30}{30}$ Training/Test, in which, for the test set, the last 30 templates of the total were used;$\frac{30}{20}$ Training/Testing, in which, for the test, the last 20 templates of the total were used;$\frac{30}{15}$ Training/Testing, in which, for the test, the last 15 templates of the total were used.

This ensures that the templates used for testing did not have the same temporal sequence as the randomly obtained ones, thus demonstrating that even if the morphology of the signal slightly changes, the classifier can still identify the subject. For both characterizations, the process was repeated 30 times so that the generalization ability of the algorithm could be more accurately evaluated. In addition, the performance of the classifier as a function of the number of subjects was analyzed. That is, the classifier was tested with templates for a number of subjects ranging from 2 to 17.

### 4.3. Classification

Different numbers of subjects were tested to simulate families of different sizes. Binary Convolutional Neural Network (BCNN), Support Vector Machine (SVM), Gaussian Naive Bayes (GaussianNB) and K-Nearest Neighbour (3-NN) classifiers were explored for thigh ECG-based identification. These classifiers were chosen because they have already been addressed in the literature as a method for identifying individuals using ECG signals [18,19,20].

Due to the novelty of the problem, a Neural Network (NN) had to be developed from the ground-up. A 1D Convolutional Neural Network (CNN) was picked as a baseline because it is the current state-of-the-art for ECG-based identification. A CNN is also very skilled at finding spatial features, which is especially attractive for time-series data, where successive datapoints are time-dependent and related to each other. In this work, we explore a novel architecture based on Binary Neural Networks (BNN), i.e., a CNN-based BNN (BCNN), which maximizes the computational performance [21]. Similarly to state-of-the-art works, we used the filtered and segmented ECG recordings (templates) as inputs. After the segmentation, the BCNN input data were represented as 32-bit floating point numbers. To minimize the memory footprint, the data were quantized to 8-bit unsigned integers. It is common practice to perform standardization on the NN’s input data, which we also adopted [21].

For the biometric identification method using SVM, we used the multiclass SVM, one-against-all (OAA) with Linear Kernel [22]. GaussianNB consists of determining a posteriori probability by multiplying the a priori probability by the probability of a positive outcome, called the conditional independence of class [18]. The 3-NN algorithm results in a classification where an object is classified by a majority vote of its neighbours, with the most frequent object among its three nearest-neighbours being assigned to the class [23].

## 5. Results

We performed signal quality assessment through rhythm and waveform morphology analysis, evaluated the thigh ECG-based identification performance, and the user satisfaction. The results for each of these experiments are presented in this section.

### 5.1. Rhythm Analysis

With this analysis, we seek to characterize potential differences in the heart rate as computed from the signals collected with ECG REF (gold standard) and with ECG EXP (Silicone Electrode). Distortions in the R peak may introduce a latency that affects the heart rate calculation, and artefacts in the signal may lead to undetected or wrongly detected peaks. Due to the influence of noise, there may be a lack of compatibility between the QRS complexes of the EXP and REF sensors. It is, therefore, important to evaluate the correlation between the heart rates and P- QRS -T morphology obtained with the EXP and REF sensors. It is worth noting that the heartrate is only performed when there are two or more consecutive R-peaks.

In this section, the statistical analysis with mean (μ) and standard deviation (σ) of the heartrate difference in the EXP sensor relative to the REF sensor is presented for each pair of consecutive R peaks. The signal detection error (SDE in %) is also examined, which is the percentage of segments that are saturated or corrupted by noise, given by Equation (Equation 2), where *S* represents the total signal and *N* the out-of-range signal (both in seconds). All the results are summarized in Table 3.
(2)$$SDE\left(\%\right)=\left(1-\frac{\left(S-N\right)}{S}\right)\times 100$$

Figure 7 shows a radar plot of Heart Rate Variability (HRV) parameters to provide a visual comparison of parameters calculated from the R-peaks in ECG EXP signals comparatively to those obtained from the ECG REF. The radar plot normalizes the values of the NNI reference series with the values extracted from the experimental series, and this series is used with the reference values at 100% (also described in Table 4).

The results show that the signals obtained from the population under study had a 75% recognition of QRS complexes, with a 15% excluded signal on average. For this new acquisition system, these are positive results, especially when considering that the acquisition is carried out at the thighs. A great similarity was found between the heartrate obtained using the ECG EXP and the ECG REF, which encourages us to further study the thighs for ECG acquisition.

### 5.2. Heartbeat Waveform Morphology

An analysis was performed to evaluate the morphological similarities between the heartbeat waveforms detected with the REF and EXP sensors. Following the methodology described in [5], the R peaks in the REF sensor were first determined and correlated with the cycles of the EXP sensor for each beat cycle. In this way, it was possible to detect and remove the outliers of the heartbeat waveform, to ensure only valid waveforms were obtained in the EXP sensor.

In Figure 8 we illustrate the individualized heartbeat waveforms, with the heartbeat waveforms that were considered valid represented in yellow and the heartbeat waves considered as outliers represented in gray. Table 5 presents the values of Pearson’s Correlation Coefficient (PCC) and Normalized Root Mean Square Error (NRMSE). Based on the results from Section 5.1, which were further confirmed experimentally, the Silicone Electrode shows the best performance compared to the PLA electrode. However, despite the high morphological correlation, there are significant differences that could influence the biometric recognition results, again reinforcing the importance of studying this topic and the work. In Figure 9 we have an example of the ECG EXP signal (green) obtained using the toilet lid compared to the signal obtained using the ECG REF (blue). The red and purple points show the R peaks detected by the algorithm used in this work.

### 5.3. Biometric Identification

Figure 10 and Figure 11 present the performance, expressed in terms of accuracy, recall and precision for each tested classifier, as a function of the population size. Figure 10 shows the results for the random template selection described in Section 4.2, which achieves an accuracy between 97 and 100% for a population of up to four subjects. As the population increases, as expected, the performance of the classifier decreases and among the four tested classifiers, the one with the best results was the BCNN. Figure 11 shows the results for the static template selection described in Section 4.2.

In these results, it is possible to observe a reduction in the performance of the classifier that, for a population with up to four subjects, presents an accuracy of 100%. The BCNN is the best classifier and proves that when values with distant temporal intervals are used, there may be a greater difficulty in correctly identifying the subject, because its morphology may be slightly different over time. However, for all cases, and based on the literature [24], it was possible to obtain viable user identification results through ECG signals collected at the thighs.

### 5.4. Satisfaction Form

Besides the functional evaluation, in this work, we also performed a user satisfaction survey as a way of assessing the subjects’ acceptance of the proposed device. This can consider the subjects’ overall experience with the device, or specific aspects of their interaction. With the survey results, it is possible to identify what needs to be improved from a user’s perspective and what can already be considered stable. With this part of the study, we seek to evaluate how users perceive that their needs are addressed by the device. Table 6 details the population that was enrolled in the tests.

To this end, the population answered a questionnaire assessing the usability of the device (Table 7) [25], and a satisfaction questionnaire about the system (Table 8). The questionnaire about the usability of the device describes the user’s opinion about the impact of the device at first sight and how comfortable it was on a scale of 1–5. The satisfaction questionnaire is used to validate the device for the end user and to assess the impact of using it. From the results, it is possible to verify the great acceptance of the device. Despite its being something that had never been seen by the users, from the System Usability Scale (SUS) responses, it is possible to calculate a usability index of 96%, allowing us to conclude that the subjects participating in the trial considered it to be easy to use and learn; 75% of the subjects would use the system frequently. Comfort is also essential for adoption of the device, with 19% of respondants finding it comfortable and 81% extremely comfortable given its intent.

## 6. Conclusions

This paper builds upon our previous work, further contributing to the state-of-the-art in invisible ECG by exploring the feasibility of signals collected at the thighs, with a sensor integrated in a toilet seat, for subject identification. Our approach enables ECG signal acquisition without the use of body-worn devices, bringing a new paradigm to automated health-monitoring systems embedded in the subjects’ living space.

Based on the experimental evaluation by the morphological analysis of the collected ECG signals, heartrate variability (HRV) and heartrate, the custom textured conductive silicone dry electrodes showed satisfactory results for the acquisition of ECG signals on a toilet seat (described in Section 5.1 and Section 5.2). A proof-of-concept of a smart toilet seat that allows for the identification of different users based on ECG recordings was created. This proof-of-concept summarises the technical solutions that have shown the best performance during development and serves as an essential theme for the expansion of this topic in the prior art (described in Section 5.3).

Future work will be focused on performing further data acquisition in healthy controls and subjects with cardiovascular diseases, evaluating if/how the biometric templates change over time (e.g., multiple months apart), exploring privacy-preserving template representations, and devising automated biometric template update strategies. In addition, we envision the study of template fusion techniques, and assessment of the minimum contact time with the device, not only for biometric identification but also to allow for the early detection of cardiovascular disorders. However, this work further strengthens the feasibility of thigh ECG data acquisition using electrodes embedded in a surface with which subjects interact on a regular basis [26].

## Figures and Tables

**Figure 1 sensors-22-04201-f001:**
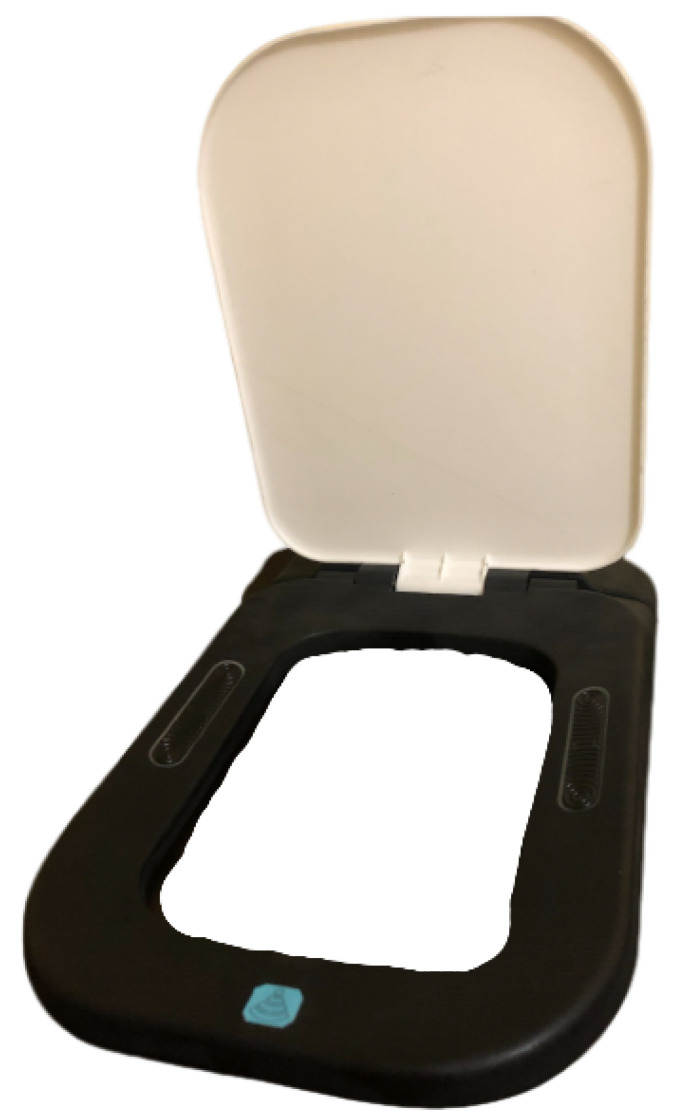
Prototype of the toilet seat, highlighting the electrodes’ positioning.

**Figure 2 sensors-22-04201-f002:**
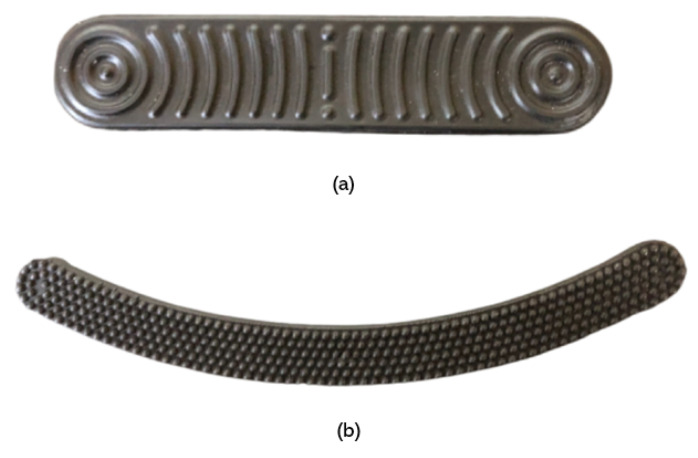
Electrode texture and geometries used in the scope of this work: (**a**) PLA electrode; and (**b**) Silicone electrode.

**Figure 3 sensors-22-04201-f003:**
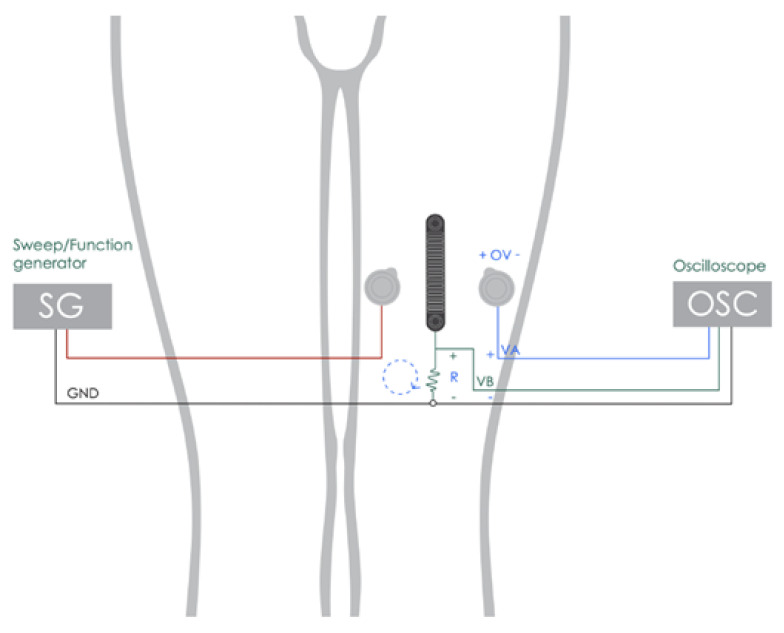
Experimental setup for skin-to-electrode impedance measurement.

**Figure 4 sensors-22-04201-f004:**
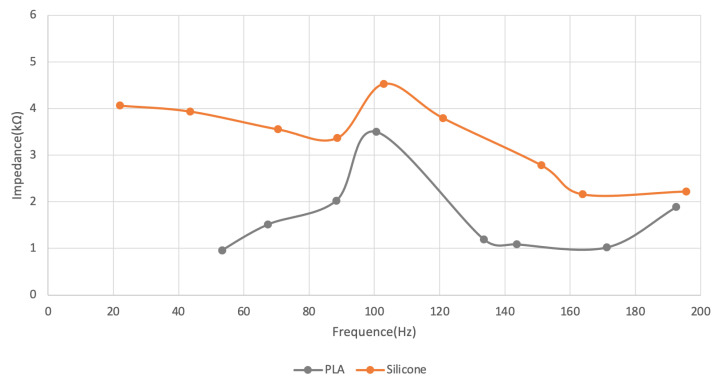
Impedance between skin and electrode.

**Figure 5 sensors-22-04201-f005:**
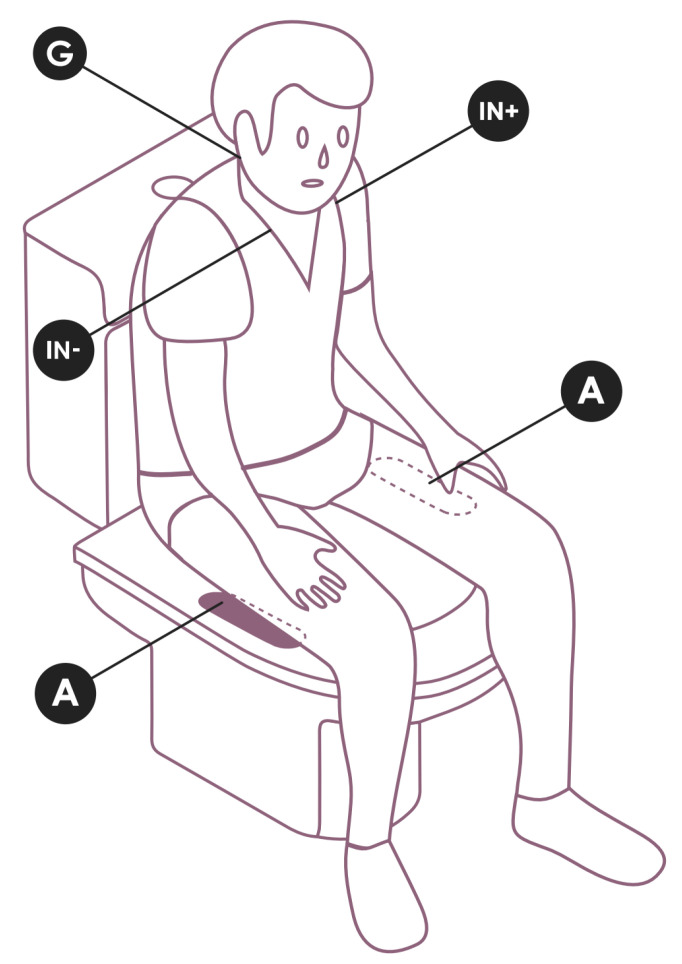
Experimental setup showing the electrode placement for the reference system (ECG REF: IN+, IN− & Ground) and the experimental electrodes location on the toilet seat (ECG EXP: A).

**Figure 6 sensors-22-04201-f006:**
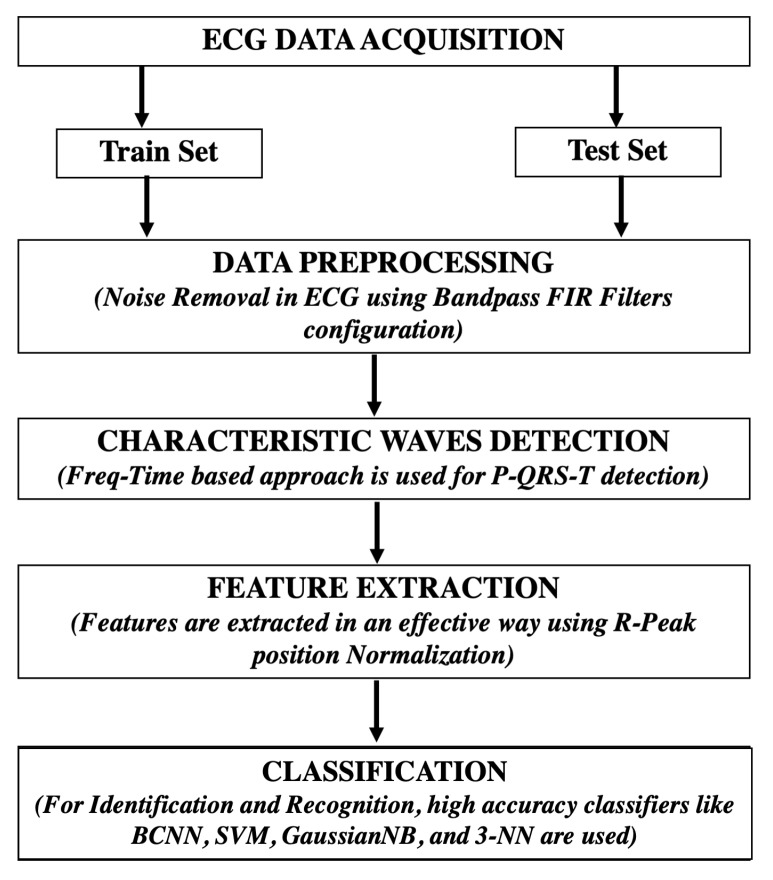
Methodology of an ECG biometric identification system.

**Figure 7 sensors-22-04201-f007:**
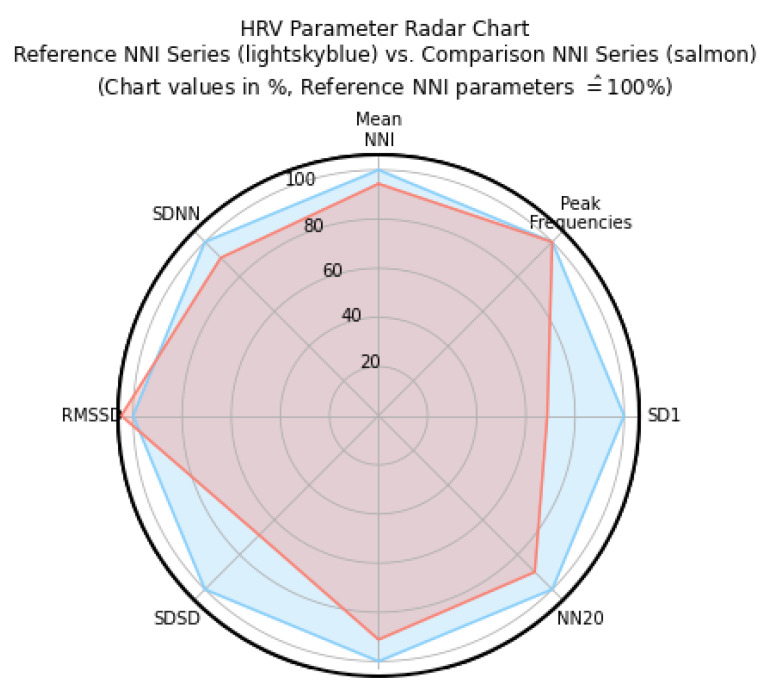
Radar plot for the reference NNI series (extracted from ECG REF) and for the comparison NNI Series (extracted from ECG EXP).

**Figure 8 sensors-22-04201-f008:**
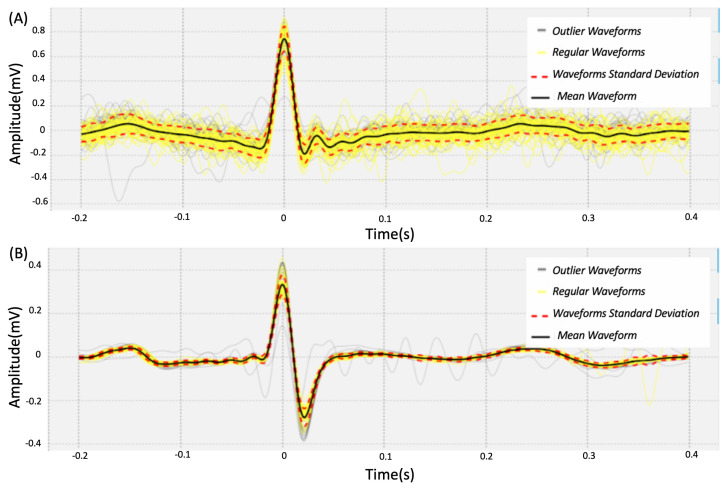
Example heartbeat waveforms for a test subject: (**A**) EXP and (**B**) REF electrodes.

**Figure 9 sensors-22-04201-f009:**
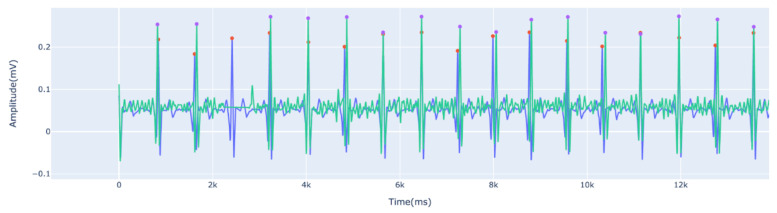
Example of the ECG EXP signal (green) obtained using the toilet seat compared to the signal obtained using the ECG REF (blue). The red and purple dots show the R-peaks detected by the algorithm used in this work.

**Figure 10 sensors-22-04201-f010:**
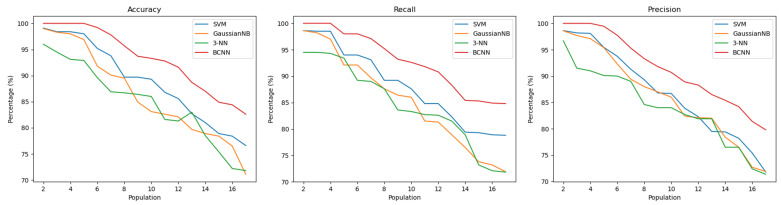
Accuracy, Recall and Precision of the SVM, GaussianNB, 3-NN and BCNN classifiers for random template selection.

**Figure 11 sensors-22-04201-f011:**
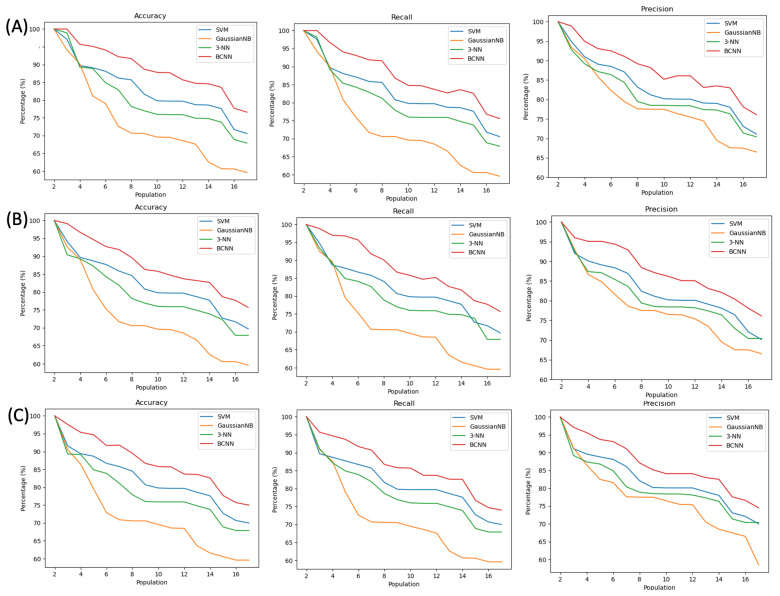
Accuracy, Recall and Precision of SVM, GaussianNB, 3-NN and BCNN classifiers for static characterization with 1/3 test (**A**), 2/9 test (**B**) and 1/6 test (**C**).

**Table 1 sensors-22-04201-t001:** Properties of the high consistency silicone rubber.

Test	Standard	Units	Typical Values			
Physical Properties						
Density	ASTM D792	g/cm^3^	1.15	1.17	1.18	1.20
Tensile Strength	ASTM D412	MPa	6.00	6.30	6.00	5.00
Elongation @ Break	ASTM D412	%	250	220	195	175
Tear Strength	ASTM D624 C	kN/m	15	12	14	13
Electrical Properties						
Volume Resistivity	ASTM D99189	Ω/cm	≤5	≤2	≤2	≤2
Shielding Effectiveness:	MIL-G-83528					
200 KHz (H Field)		dB	-	-	-	-
100 MHz (E Field)		dB	80	90	90	90
500 MHz (E Field)		dB	80	90	90	90
2 GHz (Plane Wave)		dB	60	70	70	70
10 GHz (Plane Wave)		dB	50	60	60	60

**Table 2 sensors-22-04201-t002:** Skin-electrode impedance for the silicone electrodes.

Silicone									
F (Hz)	21.98	43.56	70.42	88.62	102.89	121.10	151.20	163.90	195.60
Z (kΩ)	4.06	3.94	3.56	3.37	4.53	3.79	2.78	2.16	2.22
PLA									
F (Hz)	53.30	53.30	67.39	88.4	100.60	133.60	143.60	171.20	192.50
Z (kΩ)	0.96	0.96	1.52	2.03	7.50	1.19	1.09	1.02	1.89

F—Frequency; Z—Impedance.

**Table 3 sensors-22-04201-t003:** Comparative analysis of the heartrate values determined for the ECG time series obtained with the Silicone Electrode (EXP).

Samples	QRS (%)	HR (BPM)	SDE (%)	*p*-Value	TB (s)	TS (s)
Gender: Female						
1	72.3 ± 9.8	0.1 ± 2.3	16.6	0.96	0	40
2	73.2 ± 5.9	0.4 ± 6.3	12.5	0.93	0	40
3	71.4 ± 8.3	0.2 ± 3.1	6.6	0.97	1	15
4	86.4 ± 15.3	0.7 ± 5.9	2.8	0.89	0	5
5	76.0 ± 9.8	1.3 ± 8.4	17.5	0.78	2	40
6	76.6 ± 9.8	1.3 ± 8.4	17.5	0.99	0	120
7	75.8 ± 6.8	20. ± 1.3	10.4	0.88	15	10
8	72.3 ± 9.8	0.5 ± 0.9	17.5	0.78	40	2
9	93.2 ± 4.5	3.4 ± 1.1	0.0	0.98	0	0
10	87.8 ± 13.7	0.6 ± 1.2	10.4	0.99	20	5
11	73.1 ± 10.0	0.5 ± 2.3	22.9	0.97	0	55
12	71.5 ± 4.9	6.3 ± 3.1	12.9	0.88	1	30
13	68.0 ± 5.8	0.4 ± 4.1	1.2	0.99	0	3
*μ* ± *σ*	76.2 ± 7.1	2.5 ± 5.2	18.1 ± 24.9	0.9 ± 0.0	5.2 ± 11.3	26.4 ± 31.9
Gender: Male						
14	87.3 ± 9.8	2.3 ± 0.4	12.50	0.77	0	30
15	77.1 ± 7.9	1.5 ± 0.9	7.08	0.88	2	15
16	77.4 ± 4.5	0.7 ± 0.3	20.83	0.79	40	10
17	76.8 ± 10.8	5.8 ± 5.7	29.17	0.84	30	40
*μ* ± *σ*	82.1 ± 7.8	2.8 ± 1.9	25.3 ± 23.8	0.87 ± 0.2	39.5 ± 63.7	21.2 ± 12.7

**QRS**—Percentage of the ratio between the QRS complexes detected with the EXP sensor and the REF; **HR**—Difference between HR detected with the ECG REF and experimental sensor (ECG EXP); **SDE**—Signal Detection Error, i.e., the percentage of signals that are excluded from the original signal due to excessive noise or saturation; ***p*-value**—Two-sided *p*-value (the unpaired *t*-test used in the analysis of the data derived from the signals obtained with the ECG EXP electrodes in relation to the ECG REF electrode); **TB**—Time to first beat of the ECG EXP; **TS**—Time during which the signal was saturated of the ECG EXP.

**Table 4 sensors-22-04201-t004:** HVR Parameter Radar Chart: ECG REF and ECG EXP.

Electrode	Mean NNI (ms)	SD1 (ms)	SDNN (ms)	RMSSD (ms)	SDSD (ms)	NN20	Peak Frequencies (Hz)
REF	28,663.5 ± 2.5	334.8 ± 91.78	17257.6 ± 10.9	761.1 ± 7.9	323.3 ± 8.9	80.0 ± 2.3	0.01 ± 0.0
Silicone	27,050.8± 5.7	127.89 ± 89.0	15,679.9 ± 6.8	797.5 ± 6.2	222.3 ± 6.5	72.0 ± 1.7	0.01 ± 0.0
%	−5.6 ± 2.8	41.8 ± 11.9	−9.4 ± 1.8	4.79 ± 3.2	−31.2 ± 9.8	−10.0 ± 7.3	0.1 ± 0.0

**Mean NNI (ms)**—Computes the series of NN intervals [ms] from a series of successive R-peak locations; **SD1 (ms)**—Standard deviation along the minor axis; **SDNN (ms)**—Standard deviation of NN series parameters; **RMSSD (ms)**—Root mean square of successive differences between normal heartbeats; **SDSD (ms)**—Standard deviation of successive differences; **Peak Frequencies (Hz)**—Breaths per second.

**Table 5 sensors-22-04201-t005:** Pearson Correlation Coefficient (PCC) and Normalized Root Mean Squared Error (NRMSE) between the heartbeat waveforms of ECG REF and ECG EXP.

Subject	PCC	NRMSE
1	0.88 ± 0.15	29.26 ± 10.52
2	0.98 ± 0.03	19.86 ± 11.07
3	0.81 ± 0.14	30.34 ± 16.88
4	0.95 ± 0.30	51.65 ± 6.42
5	0.97 ± 0.25	49.89 ± 3.48
6	0.89 ± 0.30	24.66 ± 12.83
7	0.86 ± 0.29	17.45 ± 4.51
8	0.99 ± 0.12	61.34 ± 2.77
9	0.88 ± 0.15	34.21 ± 5.46
10	0.97 ± 0.15	29.78 ± 5.87
11	0.98 ± 0.12	25.65 ± 12.83
12	0.83 ± 0.04	12.45 ± 6.53
13	0.82 ± 0.06	34.65 ± 3.97
14	0.99 ± 0.00	21.45 ± 3.01
15	0.89 ± 0.04	57.65 ± 16.88
16	0.88 ± 0.15	26.16 ± 10.71
17	0.87 ± 0.29	34.07 ± 3.46
*μ* ± *σ*	0.91 ± 0.06	32.64 ± 14.06

**Table 6 sensors-22-04201-t006:** Number and main characteristics of the test population.

Age Intervals		Female	Male	Weight (Kg)	Height (cm)
Young Adults	18∼28	3	0	67.0 ± 19.0	170.2 ± 11.7
	28∼38	2	1	76.5 ± 19.9	171.8 ± 13.7
Middle-Age Adults	38∼48	2	1	68.0 ± 12.6	163.5 ± 11.6
	48∼58	2	1	73.1 ± 12.1	163.2 ± 6.2
	58∼68	3	1	74.2 ± 12.6	161.7 ± 14.3
Old Adults	68∼78	1	0	75.4 ± 5.1	169.8 ± 9.7
Total		13	4	72.4 ± 13.3	165.7 ± 11.3

**Table 7 sensors-22-04201-t007:** System Usability Scale (SUS) and number of subjects whom chose each option.

Questions	1	2	3	4	5
I think that I would like to use this system frequently	0	0	0	12	5
I found the system unnecessarily complex	7	8	0	0	0
I thought the system was easy to use	0	0	0	2	15
I think that I would need the support of a technical person to be able to use this system	10	3	3	1	0
I found the various functions in this system were well integrated	0	1	11	3	2
I thought there was too much inconsistency in this system	0	17	0	0	0
I would imagine that most people would learn to use this system very quickly	0	1	3	4	9
I found the system very cumbersome to use	0	13	4	0	0
I felt very confident using the system	0	0	2	5	10
I needed to learn a lot of things before I could get going with this system	5	12	0	0	0

1—I strongly disagree, 2—I disagree, 3—I agree moderately, 4—I agree and 5—I strongly agree.

**Table 8 sensors-22-04201-t008:** Satisfaction Questionnaire and number of subjects that chose each option.

Questions	1	2	3	4
How convenient is the use of our system?	0	0	6	11
1-Nothing convenient, 2-Not very convenient, 3-Convenient and 4-Very convenient				
How comfortable did you feel using our system?	0	4	0	13
1-Nothing comfortable, 2-Not very comfortable, 3-Comfortable and 4-Very comfortable				
Compared to our competitors, is the quality of our system better, worse or the same?	0	11	6	0
1-N/A, 2-Worst, 3-Same and 4-Better				
To what extent would you order our system?	0	0	16	1
1-Unlikely, 2-Likely, 3-Very likely, 4-Extremely likely				

## Data Availability

Not applicable.

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
