# Peer review of "Identity Recognition in Sanitary Facilities Using Invisible Electrocardiography"

_sensors, 2022, doi:10.3390/s22114201_

Round 1

Reviewer 1 Report

  1. The introduction lacks persuasiveness and reference support. The literature review is not enough. For example, the authors did not comment on the papers below (I just cited some)
  • Zixuan Zhang et al. Artificial intelligence of toilet (AI-Toilet) for an integrated health monitoring system (IHMS) using smart triboelectric pressure sensors and image sensor. Nano Energy, Volume 90, Part A, December 2021, 106517
  • Kurahashi, K. Murao, T. Terada and M. Tsukamoto, "Personal identification system based on rotation of toilet paper rolls," 2017 IEEE International Conference on Pervasive Computing and Communications Workshops (PerCom Workshops), 2017, pp. 521-526, doi: 10.1109/PERCOMW.2017.7917617.
  1. This is a very interesting manuscript. But I think it's mostly for women (women need to sit on the toilet seat when they go to the toilet.) not men (except men are pooping)
  2. I think the authors also need to touch the thigh with one or both hands to evaluate the data. This may affect the signal.
  3. As the authors mentioned in the manuscript, accuracy, recall, and precision decrease as the population increases. This means that this method is not practical for large families. A family usually consists of 4 members and the accuracy, recall and precision will drop to around 90%
  4. think the ECG signal when a person is sitting still is different from the ECG signal when they poop. I think the authors need to take this into account when testing the model.

Reviewer 2 Report

The manuscript proposes a method to recognize persons by invisible ECG signals in sanitary facilities. The experiments reveal that the method can achieve good results.
My comments on the current manuscript are the following:
1. In Introduction and Related work, some essential works are missing, such as 10.3390/s16101744, 10.3390/e18080285, 10.3390/app10144741 and others. The authors are encouraged to introduce these works.
2. The details of feature extraction should be clearly presented. How many features are extracted for classification?
3. One limitation of the current study is that it has only 17 subjects in the experiments. How to demonstrate the generality of the proposed method using such a small dataset?
4. The used classifiers, SVM, GNB and KNN, are all very old ones. If the users can use some other classifiers, such as random forests and xgboost, for comparison, it would improve the manuscript. 

Reviewer 3 Report

The authors present a new method for identity recognition on a toilet seat based on the use of polymeric electrodes, thigh ECG signals and machine learning. Their particular interest is on the performance of three different classifiers developed for the biometric identification of human subjects. After reading this manuscript, many questions remain not clearly answered:

What is the use/advantage of this rather complex biometric identification? - The scope of this work is reduced by the authors “to match the recorded data to the subject that provided it” within a household [line 58]. But there are much simpler methods to do this in case of uncertainty, e.g. assignment of data to the user’s profile by the user himself (as it is done by commercial smart scales).  

What is scientifically new? – some parts of the manuscript are similar to the previous paper [5], e.g. using the same toilet seat electronic system, the same data acquisition, the same signal pre-processing, the same outlier detection method. The basic concept is not a new approach to automated health monitoring systems because ECG measurement on a seat, chair or even toilet is being investigated for min. 20 years (and a few references should be mentioned and cited), and biometric identification from ECG signals the same, and also the use of conductive silicone as electrode material.

Is there really a relevant medical use? – This is not answered in the manuscript, although the medical use is questionable, because the proposed concept is by far not comparable to a long-term ECG acquisition, even not to a Smart Watch ECG acquisition which you can activate whenever you want and as often you want. Even if the acquisition method may be interesting from the technological point of view, just the fact that such a method is feasible is not enough.

The authors should definitely address these important questions in the revised version to improve the quality of the manuscript. They should rethink/reorganize the scope and contents of this manuscript.

The practical relevance of the investigated approach for identity recognition is questionable when looking at the results. I would encourage the authors to discuss their results much more. Furthermore, the question remains, if there is a use of such thigh ECG signals for health monitoring? If not, then there is no serious use for such a system. The fact that you can match an ECG waveform to a subject doesn’t say much about its quality. But the raw signal quality is important for medical grade health monitoring. The validity of the collected data is questionable for practical use because it should make a difference in terms of artifacts resulting from movement and muscle activity if a person is just sitting in a defined and relaxed manner on a toilet seat doing nothing (as it was in this study) or if a person is really using the toilet.

Moreover, the fact that you can almost match up to 4 subjects out of 17 may sound for some readers like a rather disappointing approach, in particular when saying “ECG has been shown to have low within-subject variability and high between-subject variability” [line 40]. The biometric identification of humans should be unique or 100% fail-safe.

And may I ask the authors about their statement regarding the competing interests? One of the authors is known as Co-Founder & Chief Innovation Officer at PLUX wireless biosignals, S.A., Lisbon, Portugal. This information is not given in the manuscript, although some commercial hardware/development kit of this company is used for the ECG data acquisition system.

Finally, some parts of the manuscript need further correction or explanation as described below:

  • Title:
    • I would suggest not to use the abbreviation ECG in the title
  • Related work:
    • Add refs at least for toilet seats, e.g. …
    • JMIR mHealth and uHealth - In-Home Cardiovascular Monitoring System for Heart Failure: Comparative Study
    • https://ieeexplore.ieee.org/document/4649664
    • https://ieeexplore.ieee.org/document/1403688
  • Experimental Setup:
    • What is the reference device?
  • Classification and Identification:
    • Line 196: … with a 30-order philtre ???…
    • Section 4.2: the authors should explain which ECG signal characteristics/features were obtained
    • There is nothing said about special requirements when using ECG waveforms for biometric identification: are their special features known from literature that are different from the ones used for health monitoring???
    • The so-called biometrics pipeline should be described with some details. The manuscript must be comprehensible without having to read the previous paper [5]. Information about feature selection, feature extraction, data labeling and segmentation is missing. And which feature table is exactly used as training input? How does the mentioned template look like?
  • Results:
    • Table 4 is in the wrong place, also the meaning of the HRV features is not explained
    • SD1 is missing in table 4, also the caption should read “HRV parameter radar chart: ECG REF and ECG EXP
    • The meaning of the last row in Table 4 is not explained, also what is the meaning of the peak frequencies and why are these values 0 Hz ???
    • NN50 is not readable in the radar plot
    • The meaning of SDE, TB and TS in Table 3 is not clear. For which ECG is it, EXP or REF?
    • Why is SDE a measure for error when the out-of-range signals N are subtracted? The result would be the percentage of the good signal? Also, if the result should give percent, then the formula is wrong.
    • The results in 5.1 are not well comprehensible and their findings are not discussed at all
    • The results in 5.2 may be useful for comparing the REF and EXP waveforms, but the practical use is questionable, because the classification is based on cleaned valid EXP waveforms. But in practice you won’t have REF waveforms to detect and remove outliers. The authors should discuss this issue.
    • The authors should add a figure showing raw signals from both sensors for a larger time window, not only one waveform.
    • Regarding biometric identification and following the continuous comparison between the REF and EXP data, it would be interesting to see the performance of the REF data for classification and identification.
    • A comparison/discussion with performance results for identification from literature is missing.
    • Table 8: there is no statement about the answer that 11 subjects think that the quality of the proposed system is worse than the competitors.
  • Conclusion:
    • Should also include all the limitations, drawbacks and practical implications of the proposed system.
  • References:
    • Please check ref list, I found the following issues:
    • 12 and 14 are the same
    • 15 should reference to Hamilton
    • 17 should reference to BioSPPy, also the DOI of the given ref there is not valid

Round 2

Reviewer 1 Report

The authors have tried to revise this manuscript and add neural network into the manuscript. It is worth for publication.

Reviewer 2 Report

The revisions improve the manuscript. I think it can be accepted now.

Reviewer 3 Report

Thank you for the revised version. My questions were partially answered satisfactorily by the authors. However, it would have been good to put some of the information given in the responses into the manuscript to improve its understanding.

Please check again:

Point 16: SD1 is still missing in table 4, also in the caption RMSSD is not described and why is the peak frequencies “breaths per second”?

Point 20: I assume that the calculated results for SDE are correct but I still think that equation (2) is wrong as it is written. The result of this equation is a number between 0 and 1 without unit, not percentage. Also, if N is the corrupted signal part then the result of this equation is the “good” signal part.

Point 21: The results in 5.1 are not well comprehensible and their findings are not discussed at all. à this point has not been answered. This section just describes the content of tables and plots but not the findings of these results.

Point 24, 25, 26, 27: the authors responded that they have incorporated these suggestions in the revised version, but I cannot find anything in the document!!! Please indicate in the text or respond that you have not considered it for any reason.

Section 5.3: this section has not been updated with the new classifier!!!

Finally, looking at response 7, I would recommend to mention the involvement of the senior author into the mentioned companies as a potential conflict of interest. Your response reveals another involvement, CardioID, because the senior author is one of the inventors of the company’s patented biometrics technology. This research is commercially driven and this is fine. But then, it should not be a problem to mention the involvement into the research and the commercial part of this work to show the situation more transparent. If a person is involved on both sides, a conflict of interest cannot be excluded. For example, the results of this and further work could be a benefit for PLUX to further improve or adapt the Bitalino for the described application and selling quantities of the custom version to CardioID. CardioID would sell quantities of the acquisition system and give license of the patented technology to OLI. Of course, this is the way technology transfer and commercialization should work. But then, it should not be a big deal to give a transparent statement about the engagement of authors.

Round 3

Reviewer 3 Report

Thank you for answering my comments. There is just a small mismatch left between Table 4 and Figure 7: SD1 is indicated in the chart but not given in the table.
